∂ | **Open Peer Review** | Applied and Industrial Microbiology | Research Article

# Unveiling the multifaceted potential of *Pseudomonas khavaziana* strain SR9: a promising biocontrol agent for wheat crown rot

Shengzhi Guo,[1] Yuqi Liu,[1] Yanling Yin,[1,2] Yating Chen,[1] Siyu Jia,[1] Tong Wu,[1] Jun Liao,[1] Xinyan Jiang,[1] Hafiz Abdul Kareem,[1] Xuejun Li,[3] Junfeng Pan,[1] Yao Wang,[1] Xihui Shen[1,2]

**ABSTRACT** *Fusarium pseudograminearum*, a soil-borne fungus, is the cause of the devastating wheat disease known as wheat crown rot (WCR). The persistence of this pathogen in the soil and crop residues contributes to the increased occurrence and severity of WCR. Therefore, developing effective strategies to prevent and manage WCR is of great importance. In this study, we isolated a bacterial strain, designated as SR9, from the stem of wheat, that exhibited potent antagonistic effects against *F. pseudograminearum*, as well as the biocontrol efficacy of SR9 on WCR was quantified at 83.99% ± 0.11%. We identified SR9 as *Pseudomonas khavaziana* and demonstrated its potential as a plant probiotic. SR9 displayed broad-spectrum antagonism against other fungal pathogens, including *Neurospora dictyophora*, *Botrytis californica*, and *Botryosphaeria dothidea*. Whole-genome sequencing analysis revealed that SR9 harbored genes encoding various cell wall-degrading enzymes, cellulases, and lipases, along with antifungal metabolites, which are responsible for its antagonistic activity. Gene knockout and quantitative PCR analyses reveal that phenazine is the essential factor for antagonism. SR9 possessed genes related to auxin synthesis, flagellar biosynthesis, biofilm adhesion, and the chemotaxis system, which play pivotal roles in plant colonization and growth promotion; we also evaluated the effects of SR9 on plant growth in wheat and *Arabidopsis*. Our findings strongly suggest that SR9 holds great promise as a biocontrol agent for WCR in sustainable agriculture.

**IMPORTANCE** The escalating prevalence of wheat crown rot, primarily attributed to *Fusarium pseudograminearum*, necessitates the development of cost-effective and eco-friendly biocontrol strategies. While plant endophytes are recognized for their biocontrol potential, reports on effective strains targeting wheat crown rot are sparse. This study introduces the *Pseudomonas khavaziana* SR9 strain as an efficacious antagonist to the wheat crown rot pathogen *Fusarium pseudograminearum*. Demonstrating a significant reduction in wheat crown rot incidence and notable plant growth promotion, SR9 emerges as a key contributor to plant health and agricultural sustainability. Our study outlines a biological approach to tackle wheat crown rot, establishing a groundwork for innovative sustainable agricultural practices.

**KEYWORDS** *Fusarium pseudograminearum*, *Pseudomonas khavaziana*, biological control, wheat crown rot, genomic analysis

Address correspondence to Junfeng Pan, panjf@nwsuaf.edu.cn, or Yao Wang, wangyao@nwsuaf.edu.cn.

Shengzhi Guo and Yuqi Liu contributed equally to this article. Author order was determined both alphabetically and in order of increasing seniority.

The authors declare no conflict of interest.

See the funding table on p. 15.

Plant fungal diseases threaten global food security and agricultural sustainability (1). Among them, wheat crown rot (WCR) caused by *Fusarium pseudograminearum* (2) is one of the most destructive diseases of wheat worldwide (3) (4). The incidence and severity of WCR are anticipated to intensify due to climatic factors (5). The disease affects the stem base and root of wheat plants, resulting in wilting, lodging, and yield loss (6). Moreover, the pathogen produces mycotoxins, such as deoxynivalenol and zearalenone,

that contaminate wheat grains and pose serious health risks to humans and animals (7) (8).

Chemical fungicides are commonly used to control WCR, but they have drawbacks such as environmental pollution, the development of resistance, and residue accumulation (9, 10, 11). Therefore, alternative strategies are needed to reduce the reliance on chemical fungicides and achieve sustainable management of the disease. Biological control, which involves the use of living organisms or their products to suppress plant pathogens, is one of the most promising approaches to plant disease management (12).

Plant endophytic microorganisms colonize the internal tissues of plants without causing any apparent harm to the host (13). They have been reported to have various beneficial effects on plant growth and health, such as nitrogen fixation (14), phosphate solubilization (15), production of plant hormones (16), production of siderophores (17), induction of systemic resistance (18), and antagonism against plant pathogens (9, 19). The extensive body of research confirming the efficacy of endophytic bacteria as plant probiotics stands in stark contrast to the limited documentation on specific endophytes that target WCR. This notable discrepancy opens a pathway for comprehensive research endeavors. Such efforts would focus on the identification and characterization of novel endophytes, ultimately facilitating the creation of groundbreaking biocontrol strategies.

In this study, we identified SR9, a *Pseudomonas khavaziana* strain isolated from wheat stems, with potent biocontrol properties against *F. pseudograminearum*. Comprehensive genome analysis and subsequent creation of phenazine-deficient mutant provided insights into the molecular basis of its antagonistic action. Evaluations of SR9's effects on wheat and *Arabidopsis* growth further highlighted its potential as a sustainable agricultural biocontrol agent. This study opens avenues for employing biological control tactics to combat WCR effectively.

## RESULTS

### Antagonistic potential of wheat endophytic strain SR9 against phytopathogenic fungi

We screened a series of endophytic bacterial isolates in wheat (*Triticum aestivum*), among which strain SR9 showed a strong antagonistic effect against *F. pseudograminearum* and significantly inhibited mycelial growth in a dual culture assay (Fig. 1), and the antagonism efficacy of SR9 was quantified at 67.10% ± 0.03%. Moreover, strain SR9 also exhibited antagonism against other fungi such as *Neurospora dictyophora*, *Botrytis californica*, and *Botryosphaeria dothidea*, which are known to cause various plant diseases (Fig. 1). These results indicated that strain SR9 has broad-spectrum antagonism.

In a greenhouse pot assay, we assessed the biocontrol efficacy of bacterial strain SR9 against WCR. The results indicated that wheat treated with SR9 exhibited a substantial decrease in the WCR disease index compared to the control group. Notably, the biocontrol efficacy of SR9 was quantified at 83.99% ± 0.11% (Fig. 2).

### Characterization and identification of SR9

The colonies of strain SR9 exhibit white, convex, smooth, and the regular edges, and the cells are rod-shaped (Fig. S1a) after incubation on Luria-Bertani (LB) agar for 24 h. It reached the stationary growth phase within 18 h at 30°C (Fig. S1b). It was able to utilize various carbon and nitrogen sources for growth, as shown in Fig. S1c and S1d, respectively. The carbon sources that supported its growth were glucose, trehalose, D-fructose, D-ribose, maltose, and xylan. The nitrogen sources that supported its growth were sodium nitrate, urea, glutamine, L-glutamic acid, ammonium sulfate, casein, tryptone, and yeast nitrogen base W/O amino acids.

By performing a phylogenomic analysis, we found that strain SR9 was closely related to *P. khavaziana* strain SWRI124[T], as they formed a distinct clade in the phylogenomic tree (Fig. 3). The genome comparison of strain SR9 and *P. khavaziana* strain SWRI124[T] confirmed their high similarity, with an average nucleotide identity of 97.71% (Fig. S2)

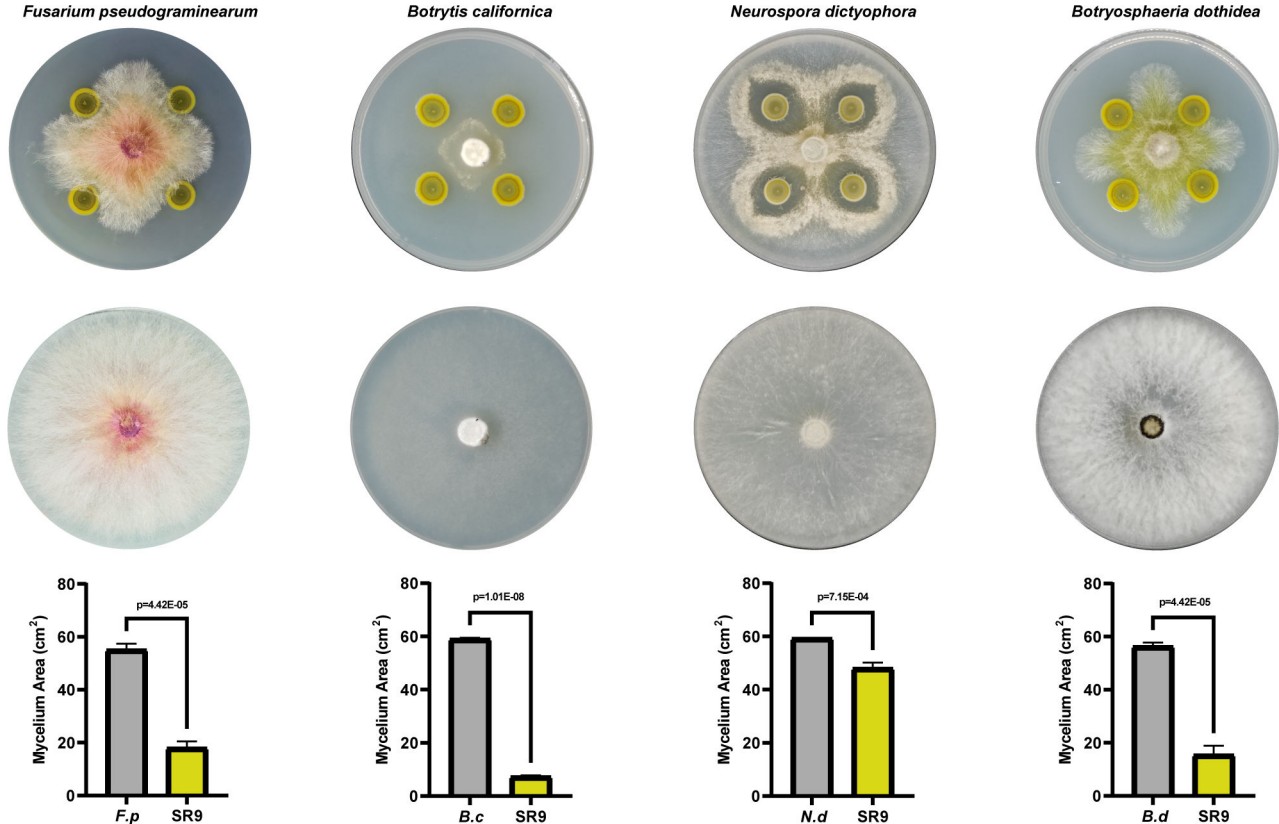

**FIG 1** Broad-spectrum antibacterial activity. Plate confrontation experiments and antimicrobial activity assessment of strain SR9 against *F. pseudograminearum*, *N. dictyophora*, *B. californica*, and *B. dothidea*. Data are mean ± SD from three biological replicates.

and a digital DNA-DNA hybridization value of 79.2% (Data S1). Based on these results, we identified strain SR9 as *P. khavaziana* strain SR9 and deposited its genome sequence in the NCBI database with the accession number GCA_036704165.1.

## Genomic features

The complete genome of strain SR9 comprises a circle chromosome, spanning a length of 6,340,526 base pairs, characterized by an average GC content of 59.91%. Genome analysis revealed the presence of 5,693 coding sequences (CDS), collectively constituting approximately 87.42% of the entire genomic content, and 107 tandem repeat regions were identified, accounting for approximately 0.26% of the overall genome. Additionally, we predicted a prophage sequence with 46,175 base pairs (location: 3515661–3561835), along with the detection of 13 gene island sequences.

## Comparative genomic, secondary metabolite analysis and antibiotic resistance

Strain SR9 was subjected to genomic comparative analysis with a panel of beneficial plant-associated *Pseudomonas* species, including *Pseudomonas mosselii* 923 (20), *Pseudomonas koreensis* UASWS1668 (21), *Pseudomonas putida* KT2440 (22), *Pseudomonas protegens* CHA0 (23), *Pseudomonas* sp. SCA7 (24), *Pseudomonas fluorescens* 2P24 (25), *Pseudomonas simiae* WCS417 (26), and *Pseudomonas mediterranea* S58 (27), as well as closely related strains in the phylogenomic tree, namely *Pseudomonas libanensis* (28) DSM17149[T], *Pseudomonas synxantha* (29) DSM 18928[T], and *P. khavaziana* SWRI124[T] (30). The image (Fig. 4a) shows the conservation and variation in gene content between genomes. The results demonstrated a close genetic relatedness among these strains and indicated that the majority of genomic regions were conserved, while the presence of

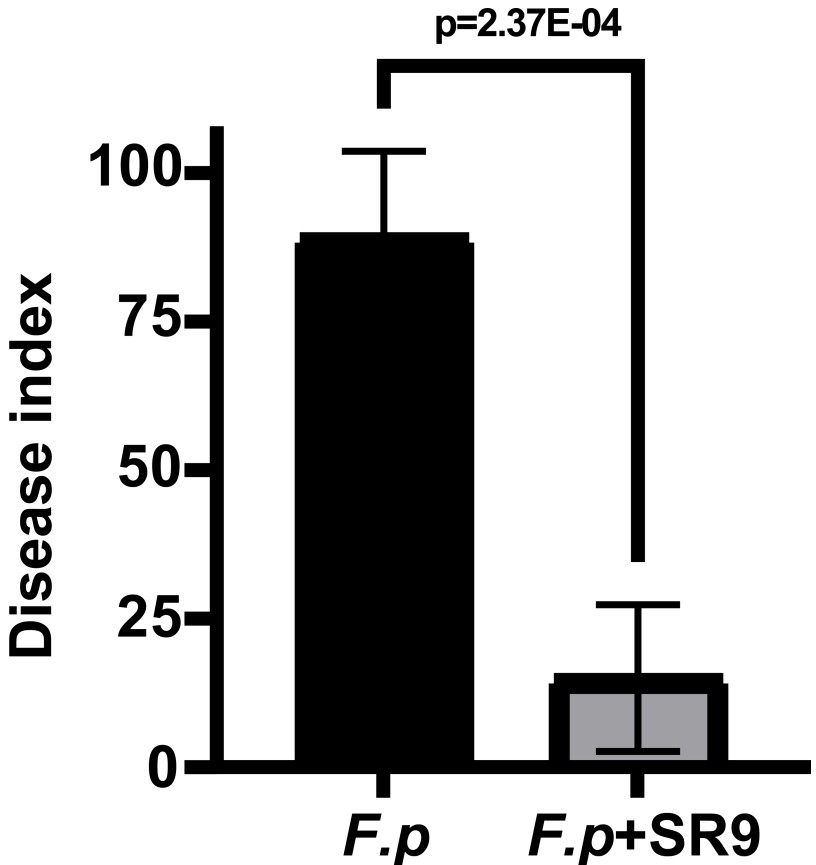

**FIG 2** Biocontrol effect of SR9 on wheat inoculated with *F. pseudograminearum* under greenhouse conditions. Data are mean ± SD from three biological replicates.

gaps underscores the distinctiveness of strain SR9. Additionally, the genomic properties of each *Pseudomonas* strains were summarized in Table S1. We selected three strains closely related to strain SR9 through a phylogenomic tree and investigated their homologous gene clusters. Our analysis unveiled the presence of a shared set of 4,601 homologous gene clusters among these strains, accounting for as much as 88.8% of all protein-coding gene orthologous clusters. A total of 4,366 gene clusters in strain SR9 were orthologous (95.1%) to *P. khavaziana* SWRI124[T]. Only 10 homologous gene clusters found in the genomes of SR9 indicated its uniqueness (Fig. 4b).

Given the antagonistic properties exhibited by SR9 against various phytopathogens, we conducted an analysis of its potential to generate secondary metabolites. This analysis predicted the involvement of 15 distinct gene clusters in the biosynthesis of various natural products. Among these clusters, 11 were non-ribosomal peptide synthetase (NRPS) clusters, responsible for the production of compounds such as azetidomonamide A, ambactin, MA026, viscosin, fengycin, and other antibacterial substances. Additionally, there were clusters responsible for the biosynthesis of siderophores like pyochelin, and pyoverdine, as well as an aryl polyene gene cluster contributing to the production of APE Vf. Furthermore, a hserlactone gene cluster played a role in endophenazine A biosynthesis, and a thiopeptide gene cluster was involved in lipopolysaccharide biosynthesis (Table 1). The prediction of these gene clusters implies that strain SR9 possesses the potential to exert biocontrol effects.

Strain SR9 has been observed to be able to thrive in the presence of ampicillin, kanamycin, nalidixic acid, chloramphenicol, and ceftazidime (Table S2). However, after conducting a genomic analysis, no resistance genes specific to SR9 were discovered. This suggests that the primary factors contributing to SR9's resistance to drugs are likely related to antibiotic efflux and alterations in antibiotic targets.

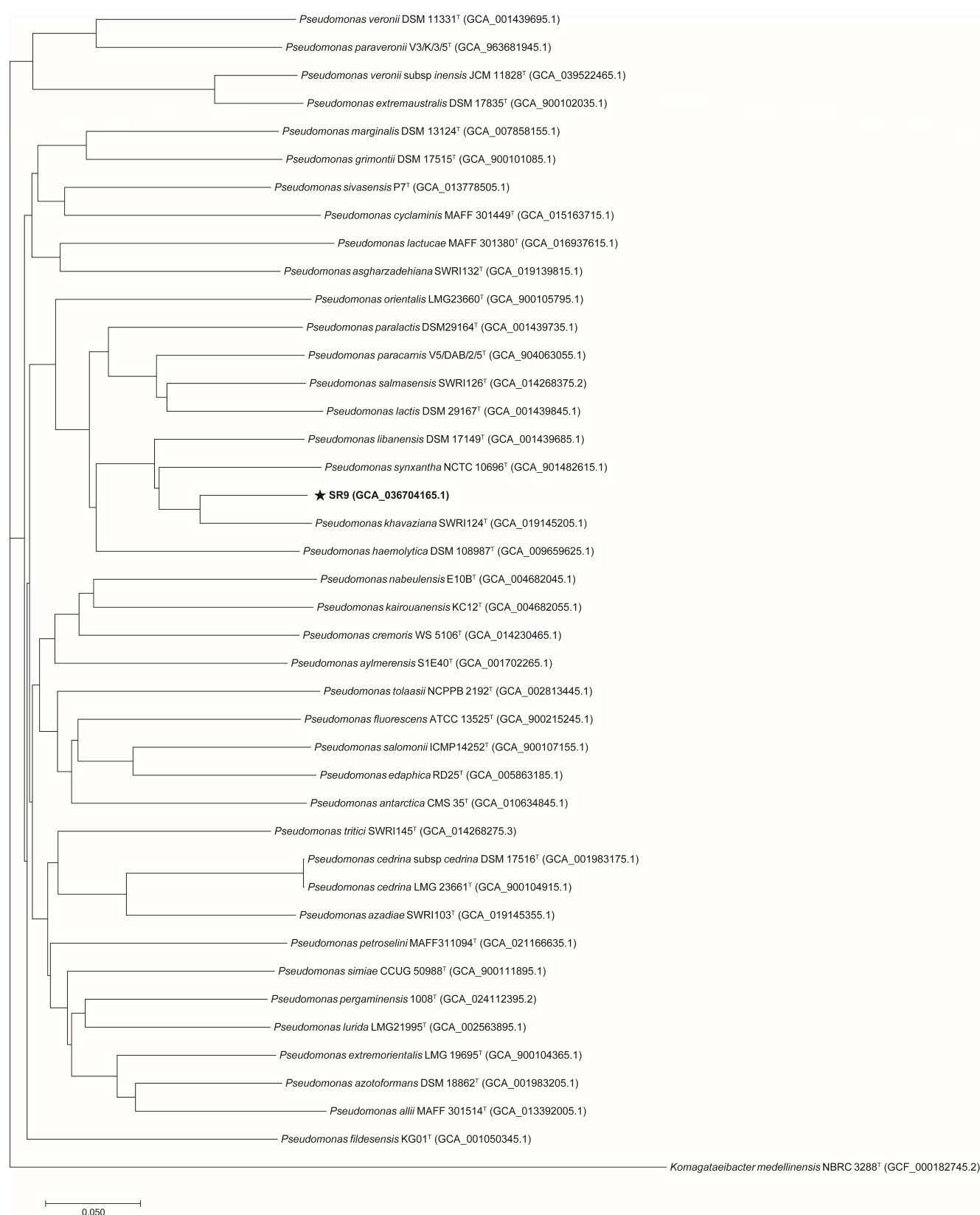

**FIG 3** Phylogenetic status of strain SR9. Phylogenomic tree based on genome sequences, showing the phylogenetic positions of strain SR9 among closely related taxa. GenBank accession numbers are given in parentheses. *Komagataeibacter medellinensis* NBRC 3288[T] were used as outgroups. [T] represents type strain.

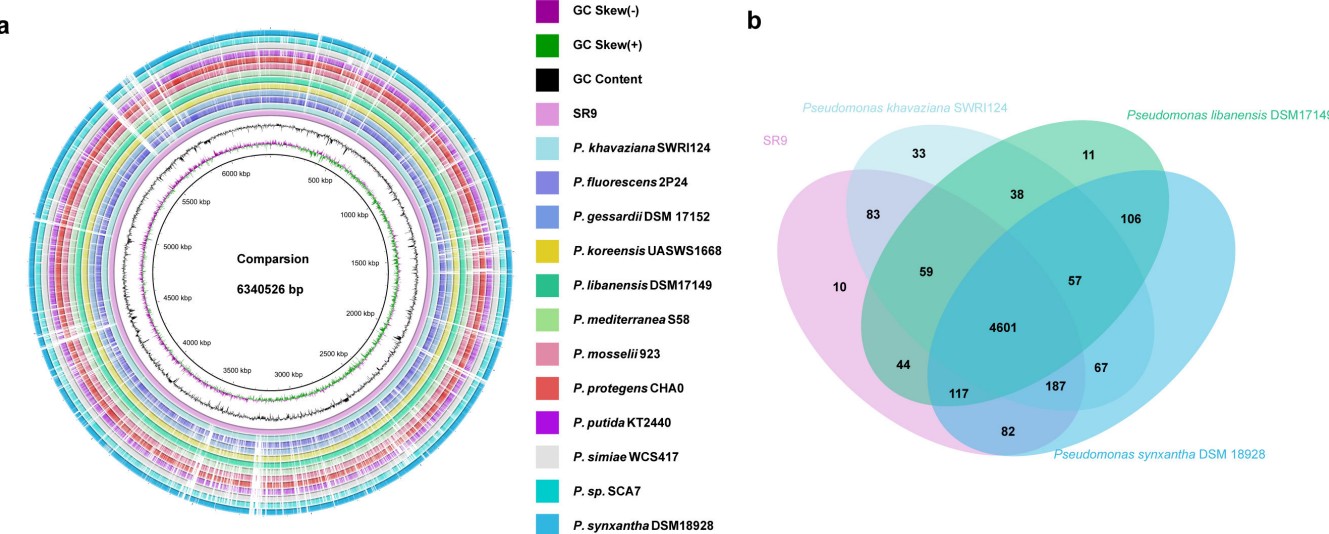

**FIG 4** Comparative genome and homologous gene cluster analysis of strain SR9. (a) Genome comparison of strain SR9 and 12 related strains. Innermost layer (pink) is a reference sequence (*P. khavaziana* SR9). The extended loops are the genomes of different *Pseudomonas* species in turn, and the colors corresponding to the specific strains are shown in the chart; (b) Venn diagram showing overlap and divergence of orthologous clusters among strain SR9 and other related strains.

## Genomic analysis of genes related to antagonistic fungi and the role of phenazines in biocontrol antagonism

We identified genes related to antagonistic fungi in the genome of strain SR9, including chitinases, cellulases, lipases, proteases, and cell wall lytic enzymes that can disrupt the fungal cell wall structure (Table S3), and secondary metabolite phenazine (Table 1) that produce antifungal effects.

To elucidate the role of phenazine in the biocontrol mechanism of strain SR9, we used homologous recombination to knock out the *phzF* gene, which encodes the key enzyme that catalyzes the conversion of phenazine precursors to phenazine, from the

**TABLE 1** List of predicted secondary metabolite biosynthetic gene clusters of *P. khavaziana* SR9

| Cluster | Type | Most similar known cluster | Similarity (%) | Gene no. | Location | Length (bp) |
|---|---|---|---|---|---|---|
| 1 | NRPS[a] | Azetidomonamide A | 100 | 36 | 191717–237569 | 45,853 |
| 2 | NRPS-like[b] | Ambactin | 25 | 21 | 325892–354595 | 28,704 |
| 3 | Aryl polyene | APE Vf | 40 | 41 | 665235–708810 | 43,576 |
| 4 | NRP-metallophore[c] ,NRPS | Pyochelin | 28 | 38 | 2593527–2657273 | 63,747 |
| 5 | NRP-metallophore, NRPS | Pyoverdine SMX-1 | 38 | 46 | 2740657–2819887 | 79,231 |
| 6 | Hserlactone, phenazine | Endophenazine A | 38 | 24 | 3345645–3368401 | 22,757 |
| 7 | NRPS, NRPS-like | MA026 | 3 | 20 | 3396585–3418615 | 22,031 |
| 8 | Thiopeptide | Lipopolysaccharide | 5 | 27 | 3611065–3637548 | 26,484 |
| 9 | NRPS | Viscosin | 43 | 39 | 3894367–3957370 | 63,004 |
| 10 | NRPS | Fengycin | 13 | 18 | 4091345–4119819 | 28,475 |
| 11 | NRPS, NRPS-like | MA026 | 14 | 36 | 4232949–4277371 | 44,423 |
| 12 | NAGGN[d] | – | – | 11 | 4513005–4527882 | 14,878 |
| 13 | NRPS | Pf-5 pyoverdine | 9 | 35 | 4569028–4621298 | 52,271 |
| 14 | NRP-metallophore, NRPS, T1PKS[e] | Secimide | 44 | 48 | 5648453–5745629 | 97,177 |
| 15 | NRP, polyketide | Lankacidin C | 13 | 17 | 5951133–5973280 | 22,148 |

[a]NRPS: Non-ribosomal peptide synthetase.
[b]NRPS-like: NRPS-like fragment.
[c]NRP-metallophore: Non-ribosomal peptide metallophores.
[d]NAGGN: N-acetylglutaminylglutamine amide.
[e]T1PKS: Type I PKS (Polyketide synthase).

phenazine biosynthetic pathway, resulting in a phenazine-deficient Δ*phzF* mutant. The primer information is listed in Table S4. We then compared the wild-type and mutant strains in terms of their antagonistic activity against *F. pseudograminearum*, as well as the effect of phenazine synthesis on the growth of the strain. We observed that the Δ*phzF* mutant strain exhibited a significant disadvantage in inhibiting the growth of *F. pseudograminearum*; restoration of the *phzF* gene successfully mitigated this inhibition deficit, culminating in an antagonistic effect on par with wild-type strain (Fig. 5a). Moreover, the comparative assessment of growth rates and biomass across the Δ*phzF* mutant, complemented, and wild-type strains revealed no significant variations (Fig. 5b). Additionally, the expression of *phzF* gene was upregulated threefold 24 h after exposure to *F. pseudograminearum* in a co-incubation assay of bacterial and fungal spore suspensions (Fig. 5c). It is noteworthy that the knockout of the *phzF* gene in the strain compromised its ability to confer resistance to wheat against *F. pseudograminearum* invasion. Complementation of the *phzF* gene markedly attenuates the incidence of WCR under greenhouse pot conditions and at the seedling stage. This genetic intervention that complements the *phzF* gene in *phzF* mutant strains results in a significant decrement in the disease index and facilitates a partial restoration of the prophylactic and therapeutic efficacy (Fig. 5d through h). Collectively, these results indicates that phenazine biosynthesis is a crucial factor in the biocontrol potential of strain SR9.

## The plant's growth-promoting characteristics

We annotated three genes associated with enzymes involved in the tryptophan-dependent biosynthesis of indole-3-acetic acid (IAA) in the genome of strain SR9, as shown in Table S5. However, the genes responsible for the initial conversion of tryptophan into intermediate compounds, such as tryptophan monooxygenase, indole-3-acetic pyruvate decarboxylase, and tryptophan aminotransferase, were not detected in the genome of strain SR9. This suggests that alternative enzymes may be involved in the initial conversion of tryptophan (31). We used the standard curve method to quantify the IAA concentration accurately, as shown in Fig. S3. The experimental data confirmed the synthesis of IAA by strain SR9. As shown in Fig. 6a and b, the concentration of IAA was $6.71 \pm 0.32$ mg/L. This finding supports our hypothesis. Moreover, we identified genes encoding essential enzymes involved in spermidine biosynthesis, as shown in Table S5.

To evaluate the probiotic effects of strain SR9 on plants, we performed growth-promotion experiments on the monocotyledonous wheat (*Triticum aestivum*) and the dicotyledonous *Arabidopsis thaliana*. Wheat plants inoculated with strain SR9 showed a significant increase in plant height and biomass compared to the control group (Fig. 6c through f). In contrast, *Arabidopsis* plants inoculated with strain SR9 exhibited a significant reduction in primary root length and a higher number of lateral roots compared to the control group. The primary root length decreased by approximately 50%, while the number of lateral roots increased by about 150% (Fig. 6g through i). In summary, these results demonstrate the remarkable ability of strain SR9 to enhance plant growth.

## DISCUSSION

Utilizing biocontrol bacteria to combat fungal pathogens and promote plant health is an increasingly vital strategy in sustainable agriculture. Biocontrol bacteria such as *Streptomyces pratensis* S10 (32), *Myxococcus xanthus* R31 (33), *Pseudomonas chlororaphis* YB-10 (34), and *Bacillus velezensis* B.BV10 (35) demonstrate effective control against various plant diseases. Further screening for novel biocontrol strains will contribute to providing new options for controlling plant diseases and promoting growth in sustainable agricultural systems. In this study, we identified the endophytic bacterium *P. khavaziana* SR9 as a potent biocontrol agent. SR9 exhibited significant antagonistic activity against *F. pseudograminearum* and successfully controlled WCR under greenhouse conditions. This is the first report of *P. khavaziana* as an endophytic biocontrol agent for

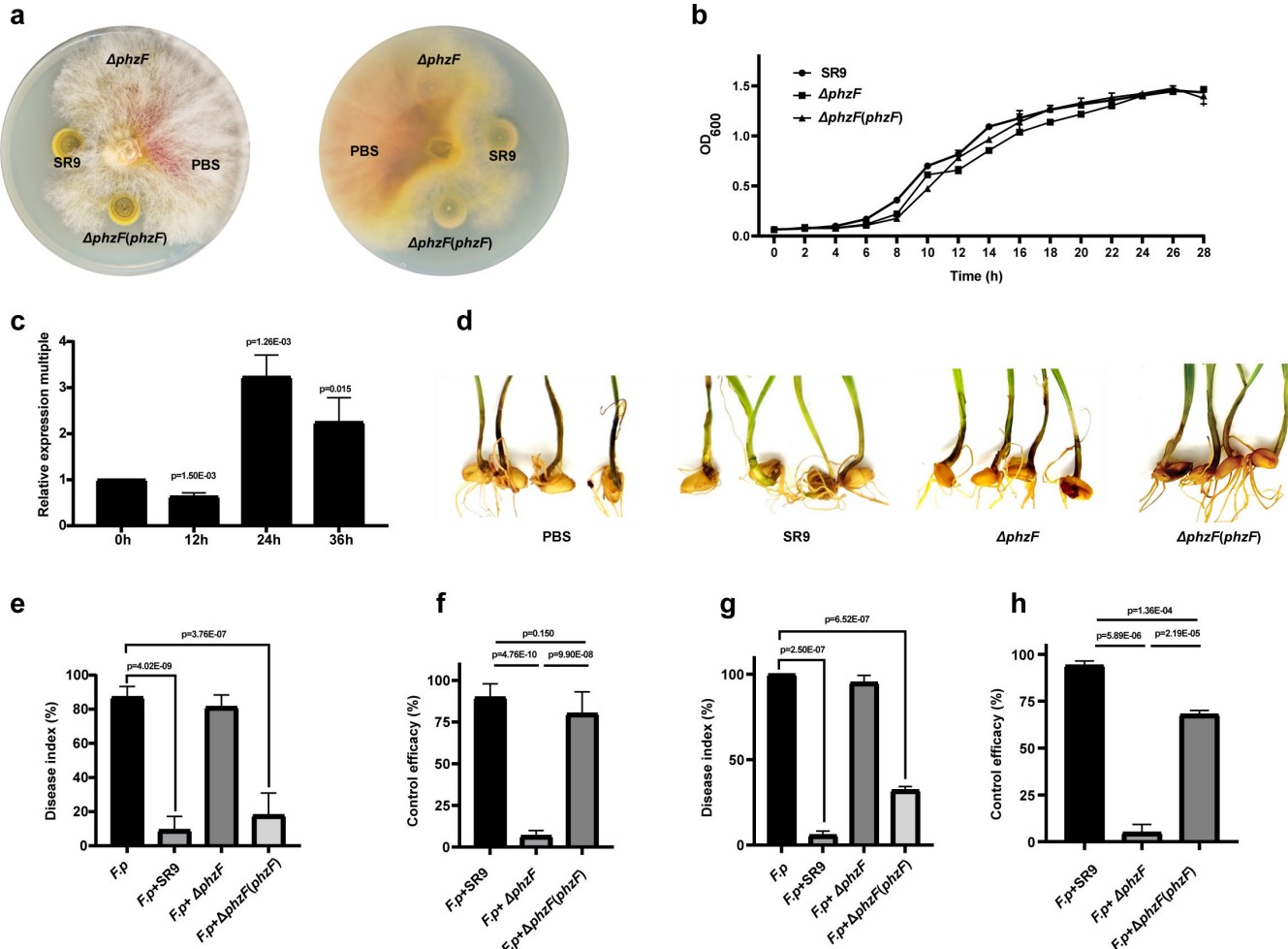

**FIG 5** The effect of phenazine on the antagonistic activity. (a) The inhibition zone diameters of wild-type, ΔphzF mutant strains and phzF complemented strains against *F. pseudograminearum* on potato dextrose agar plates; (b) the growth curves of wild-type, ΔphzF mutant strains and phzF complemented strains in LB broth; (c) fold change of phzF gene expression at different time points during co-incubation; (d) control ability of wild-type, ΔphzF mutant strains and phzF complemented strains against WCR; disease index (e) and control efficacy index (f) of SR9 and its derivatives against *F. pseudograminearum* under greenhouse potted conditions; disease index (g) and control efficacy index (h) of SR9 and its derivatives against *F. pseudograminearum* in greenhouse seedling stage. The data are presented as mean ± SD of three replicates.

wheat fungal pathogens, and one of the few reports on the biological control of WCR by endophytic bacteria.

Effective colonization is pivotal for biocontrol success. SR9 shows robust motility through flagella-driven swimming and swarming (Fig. S4), aiding in navigating the rhizosphere to locate optimal colonization sites (36, 37). Once established, biofilm adhesion creates a protective microenvironment that enhances bacterial survival and persistence under fluctuating environmental conditions (38, 39), thereby promoting plant development, nutrient uptake, and pathogen suppression (40). Genomic sequencing revealed an abundance of flagellar assembly genes and a comprehensive chemotaxis machinery (Table S6). Additionally, the biofilm-adhering capacity of SR9 significantly surpasses that of *P. fluorescens* 2P24 (Fig. S5), a known phytopathogen antagonist and plant growth promoter (25, 41).

Phenazines, nitrogen-containing heterocyclic compounds produced by various bacteria, have a broad antimicrobial spectrum (42, 43). Phenazines induce oxidative stress in fungal cells by generating reactive oxygen species, leading to cellular damage and death (44). Our study confirms the role of phenazine in SR9's biocontrol ability, with upregulation of the entire phenazine biosynthetic gene cluster (Fig. S6). The established

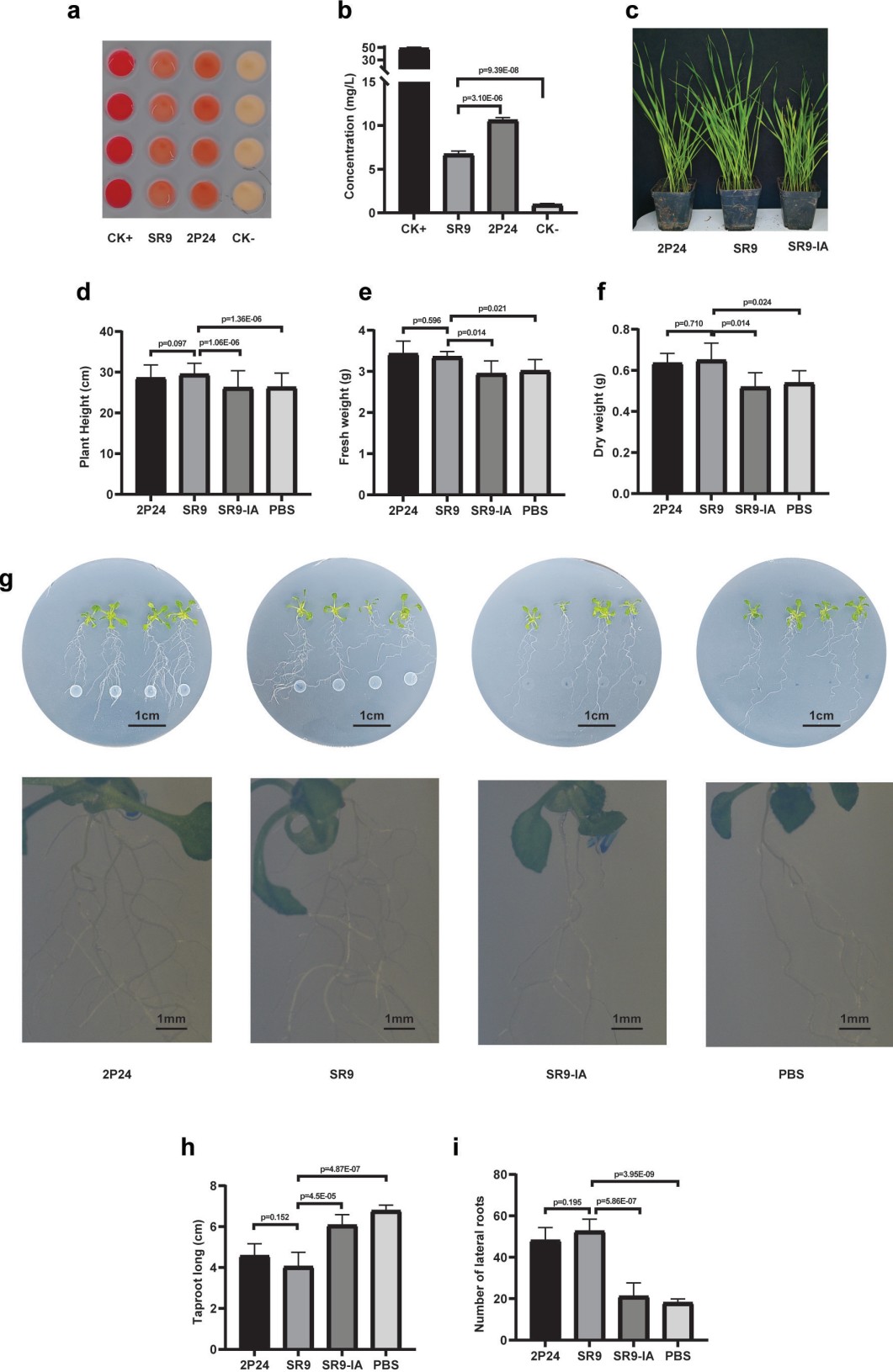

**FIG 6** Strain SR9 can produce IAA to promote plant growth. (a, b) Strain SR9 produces IAA and its quantification; CK+ represents 50 mg/L IAA, and CK– represents no bacteria suspension. Strain SR9 promotes wheat growth (c) and plant height difference (d), biomass difference (e, f); strain SR9 can shorten the main root length (i) and increase the number of lateral roots (h) in *Arabidopsis thaliana* (g). 2P24 represents *P. fluorescens* 2P24; SR9-IA represents the prepared strain SR9 suspension was inactivated.

efficacy of phenazine in the biocontrol of *Pseudomonas* notwithstanding (45, 46), its regulatory mechanisms still merit further improvement. For instance, the GacS/GacA two-component system has been confirmed to positively regulate phenazine biosynthesis (47, 48) (as substantiated by our findings, data not shown). However, the identity of the signal molecule that activates GacS/GacA (potentially originating from fungi or plants) remains elusive. Future studies should perform transcriptomic and proteomic analyses of strain SR9 under different conditions and stimuli and elucidate the molecular pathways and networks involved in its adaptation and function.

The growth and nutrition of plants are intricately linked to the symbiotic relationships they form with microorganisms in their environment (49). These relationships are particularly influential in soil, where various environmental factors come into play (50). Our research has highlighted the biocontrol strain SR9 as a key player in this symbiotic network, capable of enhancing plant growth by providing essential nutrients through solubilization of phosphate, synthesis of siderophore, and the fixation and regulation of nitrogen. Genomic analysis revealed genes integral to these processes (Table S5), confirmed by functional assays showing phosphate solubilization, siderophore production, and nitrogen fixation (Fig. S7). SR9 also synthesizes IAA, an auxin crucial for plant growth. The application of SR9 has been shown to significantly enhance the vegetative and reproductive growth phases of *Arabidopsis thaliana*, a model organism in the Brassicaceae family, as well as *Triticum aestivum*, a staple cereal crop (Fig. 6). Moreover, SR9 has a significant antagonistic effect against *B. californica* (Fig. 1), the sclerotium pathogenic fungus affecting Brassicaceae. The deployment of SR9 in crop rotation systems involving cruciferous species and gramineous crops could potentially amplify soil fertility and bolster crop yields, thereby contributing to sustainable agricultural practices.

## MATERIALS AND METHODS

### Isolation and screening of antagonism strain SR9

The healthy and thriving wheat plants from the Yangling Demonstration Zone (34.29°N and 108.07°E, China) were collected, and 3 g of stem tissue was subjected to surface disinfection (51). That is, treat with 75% (vol/vol) ethanol for 1 minute, soak in 3% (vol/vol) sodium hypochlorite for 5 minutes, then rinse with sterile distilled water three to five times, and put it on sterile filter paper to absorb the water. The surface-sterilized stems were fully grinded, serially diluted with sterile phosphate-buffered saline (PBS), and plated onto LB agar (52). The isolate, designated SR9, was obtained after incubation for 2 days at 30°C and stored at −80°C as suspensions in LB broth supplemented with 20% (vol/vol) glycerol.

### Screening of antagonism strain SR9

Mycelia of *F. pseudograminearum* were suspended in a sodium carboxymethylcellulose broth and incubated at 26°C for 5 days. Afterward, the culture underwent filtration through two layers of sterile gauze, and the resulting spores were collected via centrifugation at 3,500 rpm for 10 minutes. These spores were resuspended in sterile distilled water, with their concentration adjusted to $1 * 10^6$ conidia/mL. Subsequently, 200 µL of the spore suspension was uniformly spread onto potato dextrose agar (PDA) medium to create a screening plate.

The 37 stem endophytic bacterial strains were cultured overnight in 5 mL of LB broth and then centrifuged at 4,500 rpm for 5 minutes. The supernatant was discarded, and the bacterial pellet was re-suspended in 1 mL of sterile PBS to serve as the seed solution.

For the testing procedure, 10 µL of each seed solution was spotted onto the screening plate with 10 test strains per plate. The same procedure was repeated three times for each strain. Subsequently, the plates were incubated at 26°C, and observations were

conducted to ascertain the emergence of inhibition zones around the tested strains after 5–7 days.

## Characterization of strain SR9

Strain SR9 was routinely cultured in LB broth at 30℃ for 24 h with shaking at 180 rpm, and then harvested by centrifugation and resuspended in sterile PBS to achieve a final concentration of $1 * 10^8$ colony forming unit (CFU)/mL for further characterization and experimental procedures. Strain SR9 was observed by scanning electron microscopy (SEM) using a Nano SEM-450 instrument (FEI Co., USA). The utilization of different carbon and nitrogen sources by strain SR9 was tested by growing it in Minimal Salt Medium ($K_2HPO_4.3H_2O$ 1.5 g/L, $KH_2PO_4$ 0.5 g/L, NaCl 1 g/L, $MgSO_4.7H_2O$ 0.2 g/L) (53) supplemented with various compounds as the sole carbon or nitrogen source. When testing the carbon sources, ammonium sulfate and sodium nitrate were added as nitrogen sources, and when testing the nitrogen sources, glucose was added as a carbon source. The nitrogen-fixing activity of the strain was assessed using a nitrogen-free Ashby's medium, solidified with 2.5% agar and adjusted to pH 7.0. The medium was composed of the following per liter: 10.0 g of mannitol, 0.1 g of $CaSO_4$, 0.2 g of $KH_2PO_4$, 0.2 g of $MgSO_4$, 0.2 g of NaCl, and 5 g of $CaCO_3$ (54). Phosphate solubilization was examined using commercial Organophosphorus Medium (Haibo Co., HB8673, Qingdao, China) and Inorganic Phosphorus Medium (Haibo Co., HB8670, Qingdao, China). Siderophore production was detected using CAS Medium (Haibo Co., HB9132, Qingdao, China). Antibiotic resistance was performed by incorporating varying concentrations of antibiotics into the LB medium.

## Antagonism assay

An experiment using two types of cultures (PDA plates) was performed to evaluate the antagonism of SR9 against *F. pseudograminearum* and others fungal pathogen. The method was conducted as previously described, with minor adjustments (55). A 5 mm-diameter mycelial disk of *F. pseudograminearum* or other fungal species (*N. dictyophora*, *B. californica*, and *B. dothidea*) was placed in the center of each plate. Then, 10 µL of each bacterial inoculum was spotted 2.5 cm away from the fungal disk. Sterile PBS was used as a negative control. The plates were incubated at 26℃ for 5–7 days until the fungal growth reached the edge of the control plate. The quantification of the mycelial area was conducted utilizing the ImageJ software. The antagonism of the bacterial strains was measured by the inhibition rate of the fungal growth, which was calculated as follows: inhibition rate (%) = (control mycelial area − treatment mycelial area) / control mycelial area × 100. The experiment was repeated three times for each bacterial-fungal combination.

## Biological control pot experiment

In the greenhouse, the biocontrol efficacy of SR9 against *F. pseudograminearum* in potted wheat was assessed following a modified version of the previously described method (56).

### Preparation of inoculum

Millet seeds were sterilized at 121℃ for 30 minutes, then inoculated with fresh *F. pseudograminearum* mycelium and incubated at 26℃ for 7 days, with bi-daily shaking to ensure thorough colonization.

### Planting and treatment

Plump wheat seeds (cv. JM22) were sown in pots measuring 8.5 cm in height and 6.5 cm in diameter, with 12 seeds per pot. The pots were then placed in a growth chamber under a 12-h light/12-h dark cycle. Three days post-germination, SR9 bacterial

PBS suspension (1 * 10^8 CFU/mL) was applied at 40 mL per pot, while the control group received an equivalent volume of PBS buffer. One week after sowing, when the wheat reached the one-tiller-one-leaf stage, inoculated millet grains were buried at the base of each stem, with 10 grains per plant, followed by a layer of soil substrate. After 30 days, the incidence of WCR was surveyed, and both the disease index and the biocontrol effect were calculated. Each treatment consisted of six pots and was replicated three times.

Disease severity is determined using the following grading scale (57): 0, no disease symptoms observed in the plant; 1, root discoloration to brown or yellowing beneath the first leaf sheath; 3, first leaf sheath shows significant browning but not blackening; 7, third leaf sheath turns brown; 9, plant exhibits root rot leading to death or wilting and death of leaves.

$$\text{Disease index} = \frac{\sum \left( \begin{array}{c} \text{Each disease grade} \times \text{number of} \\ \text{infected plants of that disease grade} \end{array} \right)}{\text{The total number of plants surveyed} * 9} \times 100$$

$$\text{Control efficiency} = \frac{Disease\ index\ of\ control - disease\ index\ of\ treatment}{Disease\ index\ of\ control} \times 100\%$$

## Seedling infection assay

Based on the previous description (58), wheat seedlings (cv.JM22) germinated for 2 days were used for the seedling assay. Briefly, a wound was made at the tip of the wheat coleoptile with a needle, and then inoculated with 2 µL of bacterial PBS suspension (1 * 10^8 CFU/mL) at the wound site, while the control was inoculated with sterile PBS buffer. After 24 h, a sterile gauze containing *F. pseudograminearum* mycelial cake was wrapped around the same site. Additionally, 1 mL of the same bacterial PBS suspension was poured into the roots of the seedling tray, with the control again receiving sterile PBS buffer. The seedlings were then incubated in a greenhouse at 26°C for 10 days. The experiment was conducted in triplicate and replicated three times for robustness.

## Genome sequencing, assembly, and analysis

Genomic DNA was extracted with a commercial TIANamp Bacteria DNA Kit (Tiangen Biotech Co., Beijing, China). Whole-genome sequencing was conducted by Magigene (Guangdong), employing Illumina and Nanopore technologies. Subsequent data, encompassing reads from both second- and third-generation sequences, were processed for assembly using Unicycler v.0.4.8 (59). Gene prediction was executed through Glimmer 3 (60). Non-coding tRNA and rRNA were identified through the application of tRNAscan-SE 2.0 (61) and rRNAmmer -1.2 (62), respectively. For sRNA prediction, initial alignment to the Rfam database (63) was followed by further refinement via the cmsearch program (64). The presence of prophages was assessed via the PHAST software (65). Genome sequence annotation was performed through the use of RAST (http://rast.nmpdr.org/) (66, 67). The prediction of antibiotic resistance and carbohydrate-active enzymes drew upon resources such as the Comprehensive Antibiotic Resistance Database (68) and the Carbohydrate-Active Enzymes Database (69). The identification of secondary metabolite biosynthetic gene clusters was accomplished through antiSMASH 7.0 (70). The creation of the Venn diagram depicting the analysis of homologous gene clusters was undertaken using OrthoVenn 3 (71). The construction of a phylogenomic tree based on complete genomes was reconstructed using the CVTree v.4 method (72). FastANI (73) was employed to determine the average nucleotide identity among selected species within the *Pseudomonas* genus. Digital DNA-DNA hybridization values were calculated with the Genome-to-Genome Distance Calculator 3.0 (74). Comparisons of the genomes of specific strains within the *Pseudomonas* genus were visually presented by BRIG 0.95 (75).

## Construction of *phzF* gene deletion and complementation

The *phzF* gene was deleted from strain SR9 using a pK18*mobsacB* vector and a conjugation-mediated recombination system. The upstream and downstream regions of the *phzF* gene were fused to the pk18*mobsacB* vector by seamless cloning. The recombinant plasmid was transferred into strain SR9 by conjugation with *Escherichia coli* S17-1 λpir and selected on LB plates containing 100 µg/mL ampicillin (to which strain SR9 is intrinsically resistant) and kanamycin (conferring resistance through the pk18*mobsacB* vector). The plasmid was integrated into the SR9 chromosome by single crossover. The plasmid was excised from the chromosome by double crossover in LB broth containing 15% sucrose, which selected for the loss of *sacB*. The *phzF* deletion mutants were confirmed by kanamycin sensitivity and PCR sequencing(76) (77). To generate the complemented strains, the *phzF* gene was amplified via PCR using strain SR9 as the template. The purified gene product was ligated into the vector pBBR1-MCS5 through a double digestion method. These constructs were then transformed into the *phzF* mutant strain of SR9 to produce the corresponding complemented strains. Additionally, the empty vector pBBR1-MCS5, without any exogenous gene insertion, was transformed into the *phzF* mutant and the SR9 wild-type strains for subsequent phenotypic experiments (78, 79). Primer sequences are listed in Table S4.

## Quantitative real-time PCR of *phzf* gene

The strain SR9 cultured overnight was inoculated into Potato Dextrose Broth (PDB) at 1% (vol/vol) and incubated at 180 rpm for 12 h. Then, fungal spore suspension with a final concentration of $1 * 10^6$ conidia/mL was added. At 0 h, 12 h, 24 h, and 36 h after adding the spore suspension, 1 mL of culture was collected, and RNA was extracted and reverse transcribed for quantitative real-time PCR. The group without spore suspension was the control group, and each group had three replicates.

Total RNA was extracted and purified from bacterial samples using the RNAprep Pure kit (DP430, Tiangen) and reverse transcribed to cDNA using the reverse transcription kit (AH311-02; Transgene). qPCR was performed on a LightCycler 96 system (Roche) with the SYBR Fast kit (KK4601; Kapa Biosystems). The 16S rRNA gene served as a reference gene for normalization. Primer sequences are listed in Table S4.

## Measurement of IAA production

The production of the plant hormone IAA by strain SR9 was quantified by using a colorimetric method based on the Salkowski reagent (80). Strain SR9 was grown in King B broth (81) containing 0.1 g/L of tryptophan as a precursor for IAA synthesis. Following cultivation, the bacterial cells were removed by centrifugation at 12,000 rpm for 10 minutes, and 500 µL of the supernatant was mixed with 500 µL of Salkowski reagent, which was prepared by dissolving 2 mL of 0.5 M $FeCl_3 \cdot 6H_2O$ in 98 mL of 30% perchloric acid. The resulting solution was left in the dark for 30 minutes, and then absorbance at 530 nm was measured. The IAA concentration was calculated from the standard curve equation obtained from known concentrations of IAA.

## Plant growth promotion experiment

The effects of strain SR9 on plant growth were evaluated in wheat (*Triticum aestivum*) and *Arabidopsis thaliana*. Wheat seeds (cv.JM22) were surface-sterilized as described above and germinated in the dark at 26°C for 2 days. Uniform seedlings were transplanted into a sterile soil substrate containing 20% perlite and 20% vermiculite and then placed in a plant growth chamber with a 12-h light/12-h dark cycle. On the 7th day after planting, the wheat plants were root-irrigated with 40 mL of strain SR9 PBS suspension ($1 * 10^8$ CFU/mL), while the control plants were drenched with an equal volume of sterile water. Plant height and biomass were measured after 30 days of inoculation. Each pot

contained 12 wheat seedlings, and each treatment group had six pots (8.5 cm high ×
6.5 cm diameter). We repeated the experiment independently three times.

   *Arabidopsis* seeds (*Col-0*) were surface-sterilized and sown on Murashige and Skoog
(MS) agar plates (82). After a 3-day cold stratification at 4℃, the dishes were transferred
to the same growth chamber as the wheat plants. When the *Arabidopsis* seedlings
reached a consistent growth stage, they were carefully transplanted to fresh MS agar
plates with an equal number of plants per dish. Each seedling was inoculated with 10 µL
of strain SR9 PBS suspension (1 * $10^8$ CFU/mL) at 5 cm from the base, while the control
seedlings were treated with 10 µL of sterile PBS (83). Ten days later, the primary root
length and the number of lateral roots were recorded. *Arabidopsis* roots was observed
by fluorescence stereomicroscope using a SMZ25 instrument (Nikon Co., Japan). In this
experiment, each treatment was conducted with 12 replicates for three independent
experiments.

## Biofilm adhesion assay

The biofilm adhesion capability of strain SR9 was assessed using a crystal violet staining
method (84). Briefly, strain SR9 was cultured overnight in Tryptic Soy Broth (TSB), and
170 µL of bacterial suspension, which was diluted to an $OD_{600}$ (the optical density of the
bacterial suspension measured at a wavelength of 600 nm) of 0.05, was transferred into a
96-well microtiter plate. The plate was then incubated at 30℃ for 48 h without agitation.
After incubation, the supernatant was discarded, and the wells were gently rinsed three
times with PBS to remove the planktonic cells. The biofilm cells adhered to the wells were
stained with 0.1% (wt/vol) crystal violet for 30 minutes and then washed three times with
PBS. The excess stain was dissolved with 200 µL of 95% ethanol and the absorbance of
the solution was measured at 590 nm.

## Statistical analysis

Statistical analysis was performed using Graph Pad Prism software (GraphPad Prism 9),
and all experiments used unpaired analysis, a two-tailed Student's *t*-test, and $P < 0.05$
was considered statistically significant.

## ACKNOWLEDGMENTS

We thank Life Science Research Core Services, Northwest A&F University (Ke-rang Huang)
for technical support. We also thank the Institute of Microbiology, Chinese Academy
of Sciences (Qun Han and Rui Cui) and the Nanjing Soil Research Institute, Chinese
Academy of Sciences (Ke-huan Wang) for their help in data analysis.

   This work was supported by the grant of the National Natural Science Foundation of
China (32330004 to X.S.).

## AUTHOR AFFILIATIONS

[1]State Key Laboratory for Crop Stress Resistance and High-Efficiency Production, Shaanxi
Key Laboratory of Agricultural and Environmental Microbiology, College of Life Sciences,
Northwest A&F University, Xianyang, Shaanxi, China
[2]Xinjiang Production and Construction Crops Key Laboratory of Protection and
Utilization of Biological Resources in Tarim Basin, College of Life Sciences, Tarim
University, Xinjiang, China
[3]State Key Laboratory for Crop Stress Resistance and High-Efficiency Production, College
of Agronomy, Northwest A&F University, Xianyang, Shaanxi, China

## AUTHOR ORCIDs

Shengzhi Guo ⓘ http://orcid.org/0000-0002-0710-131X
Junfeng Pan ⓘ http://orcid.org/0000-0001-8666-4446
Yao Wang ⓘ http://orcid.org/0000-0002-7149-4234

Xihui Shen ⓘ http://orcid.org/0000-0001-6867-8887

## FUNDING

| Funder | Grant(s) | Author(s) |
|---|---|---|
| MOST | National Natural Science Foundation of China (NSFC) | 32330004 | Xihui Shen |

## AUTHOR CONTRIBUTIONS

Shengzhi Guo, Formal analysis, Investigation, Methodology, Validation, Visualization, Writing – original draft, Writing – review and editing | Yuqi Liu, Investigation, Methodology, Validation, Visualization, Writing – original draft | Yanling Yin, Investigation, Methodology, Validation, Writing – review and editing | Yating Chen, Investigation, Validation, Writing – review and editing | Siyu Jia, Investigation, Validation, Writing – review and editing | Tong Wu, Methodology, Writing – review and editing | Jun Liao, Investigation, Validation, Writing – review and editing | Xinyan Jiang, Investigation, Validation, Writing – review and editing | Hafiz Abdul Kareem, Investigation, Writing – review and editing | Xuejun Li, Methodology, Resources, Writing – review and editing | Junfeng Pan, Funding acquisition, Methodology, Resources, Writing – review and editing | Yao Wang, Data curation, Formal analysis, Funding acquisition, Methodology, Resources, Writing – review and editing | Xihui Shen, Conceptualization, Data curation, Formal analysis, Funding acquisition, Project administration, Writing – review and editing

## DATA AVAILABILITY

The 16S rRNA gene sequence of strain SR9 has been deposited in the GenBank/EMBL/DDBJ databases under the accession number OR290068. The whole-genome sequence of strain SR9 has been deposited in the GenBank/EMBL/DDBJ databases under the accession number GCA_036704165.1. Strain SR9 (CGMCC 1.62143) was deposited in China General Microbiological Culture Collection Center (CGMCC).

## ADDITIONAL FILES

The following material is available online.

### Supplemental Material

**Supplemental figures and tables (Spectrum00712-24-s0001.docx).** Fig. S1 to S7; Tables S1 to S6.
**Supplemental material (Spectrum00712-24-s0002.xlsx).** Comparison of ANI and Dddh among strain SR9 and standard strains of *Pseudomonas* genus.

### Open Peer Review

**PEER REVIEW HISTORY (review-history.pdf).** An accounting of the reviewer comments and feedback.

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
