## [Reviewer comments · Microbiology Spectrum]

Microbiology Spectrum

Unveiling the multifaceted potential of *Pseudomonas khavaziana* strain SR9: a promising biocontrol agent for Wheat Crown Rot

Shengzhi Guo, Yuqi Liu, Yanling Yin, Yating Chen, Siyu Jia, Tong Wu, Jun Liao, Xinyan Jiang, Hafiz Kareem, Xuejun Li, Junfeng Pan, Yao Wang, and Xihui Shen

Corresponding Author(s): Yao Wang, Northwest A&F University College of Life Sciences

Review Timeline:

Submission Date:	March 19, 2024
Editorial Decision:	May 19, 2024
Revision Received:	July 11, 2024
Accepted:	July 12, 2024

Editor: Frédérique Reverchon

Reviewer(s): Disclosure of reviewer identity is with reference to reviewer comments included in decision letter(s). The following individuals involved in review of your submission have agreed to reveal their identity: Sandra Tienda (Reviewer #2)

Transaction Report:

DOI: <https://doi.org/10.1128/spectrum.00712-24>

Re: Spectrum00712-24 (Unveiling the multifaceted potential of *Pseudomonas khavaziana* strain SR9: a promising biocontrol agent for Wheat Crown Rot)

Dear Prof. Yao Wang:

Thank you for the privilege of reviewing your work. Below you will find my comments, instructions from the Spectrum editorial office, and the reviewer comments.

I have now received the comments made by two independent reviewers on your manuscript. Both reviewers had contrasting opinions but I am ready to assess a revised version of your manuscript, since "novelty" is not a criterion for rejection at Spectrum.

Please attend the concerns raised by the reviewers in terms of Method description, control implementation and support of the main conclusions.

Revision Guidelines

Sincerely,
Frédérique Reverchon
Editor
Microbiology Spectrum

Reviewer #1 (Comments for the Author):

I appreciate the opportunity to review the manuscript. In their work, Guo et al. present a compelling narrative regarding the *Pseudomonas khavaziana* strain SR9 from wheat stem demonstrating its ability to control fungal pathogen causing wheat disease known as wheat crown rot (WCR). From physiological and genomic aspect, the capacity of this strain were identified. Genes involved in secondary metabolites synthesis were identified in this study. This topic is interesting and fall in the scope of Microbiology Spectrum. However, there are some flaws in the design need to be addressed before it can be considered for publication.

Although the loss of function in fungal resistance was observed in the knock-out mutant of gene *phzF*, the rationale for selecting this gene as a focus on key gene for antagonistic activity was not described in the manuscript. Whether the expression of *phzF* gene was induced facing pathogen infection could give some clues, comparing to other genes, especially other secondary metabolites synthesis genes in SR9 genome. Moreover, it is too early to draw a conclusion that phenazine is the critical factor for the ability to resist fungal disease, before the link between phenazine production in SR9 and the inhibition to pathogenic fungi is built.

Reviewer #2 (Comments for the Author):

Under my point of view, this is the beginning of a nice manuscript, it is well written, but experiments lack appropriate controls for interpretation. It is recommended to give more details in some points and to complete this with new experiments.

The work of Shengzhi Guo et al. describes the isolation and characterization of bacteria *Pseudomonas khavaziana* strain SR9, coming from stem of wheat samples from China. In vitro screening with antagonism against *Fusarium pseudograminearum*, a fungus causing wheat crown rot (WCR), allowed the authors to select this bacterium, subsequently further characterization was performed, including PGPR-related activities.

Under my point of view, this is the beginning of a nice manuscript, it is well written with some easy experiments. It has a classical hypothesis for the work, but fails arising up relevant results and to my opinion it is not novel enough to be published in this journal. In general, experiments lack appropriate controls for interpretation. It is recommended to give more details in some points and to complete this with new experiments.

Major comments:

- *Fusarium pseudograminearum* is a soil-borne fungus that cause of wheat crown rot (WCR). The biocontrol assays of this disease are carried out by inoculating *P. khavaziana* SR9 and the pathogen in the soil. And the plant growth promotion activity is carried out root-irrigated with SR9. Why are potential biocontrol strains sought in stem samples and not in soil or roots samples?
- In materials and methods, you do not mention the Ashby's medium, nor the media used in Figure 6, please include a section about this in materials and methods.
- Erroneous assays that do not allow reliable results to be obtained, for example the figure 1, the distance of inoculation of the bacteria against the different pathogens are not at the same distance, I recommend repeating the antagonism assay as in Figure 5a.
- Lack of controls in the assays, e.g., in Figure 6 at least one negative control (bacteria with no activity) and one positive control (bacteria with assayed activity) must be used for each of the assays. The same for the biocontrol assays, no uninoculated plant controls and a positive control (another bacterium with known biocontrol in the pathosystem) are not applied.
- In my opinion, determining the species *Pseudomonas khavaziana* is risky, I suggest to put *Pseudomonas* spp., complementary tests should be performed for this determination.
- Some figures are out of order, e.g., Figure 7, letters are out of order both in figure legend and in the figure.
- In my opinion, the swimming and biofilm tests do not add anything to the work.
- The discussion consists of few references (7), I think that much of it should be rewritten, focusing on discussing the wide variety of mechanisms and potential characteristics that *Pseudomonas khavaziana* strain SR9 has to be a good biocontrol and plant growth-promoting bacterium, and fails arising up relevant results.

Minor comments

- Figure 1, in the case of antagonism against *N.dictyophora* and *B.dothidea*, the growth of the fungus is irregular. How did you measure the diameter? I think that the result shown in the graph is not representative of the image. SR9 strain inhibit *F. pseudograminearum* better than *B.dothidea*, however the growth measurement is similar. I suggest that it would be more correct in this case to measure the growth area of the fungus.
- Line 91, Please, change "antifungal activity" to "antagonism or inhibition of growth", the fungal inhibition growth could be due to different factors, not only the production of antifungal compounds.
- Line 171, eliminate the word "infection". In assays of Fig. 5a, only growth inhibition is observed against *F. pseudograminearum*, not infection.
- Line 190-195, I suggest changing this paragraph from materials and methods to discussion, to discuss the importance of participation in the nitrogen and sulphur cycle for potential bacterial biocontrol and plant growth promotion. The same for line 201-203.
- Line 303, The use of the word "Biocontrol" should be replaced by "antagonism", since the screening is with a plate antagonism assay.
- Line 306, Please, indicate the number of isolates that are tested in the screening
- Figure S1, I suggest that at least the growth curve of the SR9 strain in LB include bacterial counts (cfu/ml), not just optical density measurements.

-Figure 3, Although the comparison of the genome of the SR9 strain and the SWRI124T strain of *P. khavaziana* confirmed their high similarity, with an average nucleotide identity of 97.73%, I suggest that more strains be included in the phylogenetic tree.

-Figure 5d, I suggest including, along with the photos, a graph with the disease control data, to observe if there are significant differences between the *phzF* mutant and the wild strain. It may include disease index at the end of the trial, or the area of progress under the disease curve.

-Line 327: Please, change "Antifungal Activity Assay" to "Antagonism Assay", With this assay, the antagonistic capacity against the pathogen is observed, not the antifungal activity. The same for line 329 and 335.

-Line 337: You mention that the inhibition ratio was calculated, however in the graph of Figure 1 show mycelium diameter in cm. Change it in materials and methods

-Figure 7g, please, show a closer photo, to observe the increase in the lateral roots.

-Figure 7b, e and f, I suggest using the same type of graph as in 7d, h and I, showing with dots, all the replicas.

-Line 443-445, "Each seedling was inoculated with 10 μ l of strain SR9 suspension (1×10^8 CFU/ml) at 5 cm from the base, while the control seedlings were treated with 10 μ l of sterile PBS". Bacterial suspension in what is it? in culture medium? sterile PBS? I think it is important that you indicate this.

-Line 556, I suggest to change the name of this section by biofilm adhesion.

We wish to begin by thanking the Editor and Reviewers for their very supportive and constructive comments. Here, we provide a detailed response to each reviewer's comments and the corresponding changes made to the revised manuscript. Please note the following: Text in **black** font corresponds to the reviewers' comments, and **blue** font text is our response to the reviewers' comments. In the "Marked Up Manuscript File" provided, **green highlights** reflect changes based on Reviewer 1's feedback, **yellow highlights** indicate modifications made in response to Reviewer 2's comments, and **cyan highlights** denote the comprehensive re-examination and subsequent revisions of the entire manuscript following the collective remarks from both reviewers and the editor. We hope that our revisions have satisfactorily addressed your concerns and have added value to the overall work.

Reviewers 1:

Although the loss of function in fungal resistance was observed in the knock-out mutant of gene *phzF*, the rationale for selecting this gene as a focus on key gene for antagonistic activity was not described in the manuscript. Whether the expression of *phzF* gene was induced facing pathogen infection could give some clues, comparing to other genes, especially other secondary metabolites synthesis genes in SR9 genome. Moreover, it is too early to draw a conclusion that phenazine is the critical factor for the ability to resist fungal disease, before the link between phenazine production in SR9 and the inhibition to pathogenic fungi is built.

Thank you for your insightful comments and suggestions. In the face of *Fusarium pseudograminearum* challenge, we assessed the expression of the entire phenazine gene cluster in strain SR9, which revealed an upregulation trend across all seven genes within the cluster (Fig. S6).

Fig. S6 Fold expression of each gene in the phenazine gene cluster at 24 h during co-culture.

Initially, we embarked on gene knockout editing within this cluster and, prior to the submission of our manuscript, had successfully constructed only the *phzF* mutant strain. However, as of the current juncture, we have successfully engineered mutant strains for both *phzE* and *phzF*, and have also accomplished the construction of a complemented strain for *phzF* (regrettably, the complementation for *phzE* was not successful). Subsequent assays with these mutant and complemented strains, including antagonistic plate assays and seedling infection experiments, demonstrated that neither the *phzE* nor *phzF* mutants were capable of inhibiting the growth of *F. pseudograminearum*.

(This data is not shown in the manuscript, but is only used to demonstrate that knockout of *phzE* can also result in loss of antagonism activity.)

Notably, the inhibitory effect was reinstated upon the complementation of the *phzF* gene in the *phzF* mutant background (Fig. 5a). The greenhouse experiments corroborated this finding (Fig. 5e-f). With the establishment of the *phzF* complemented strain and the phenotypic assays conducted, we posit that phenazine is a crucial factor in strain SR9's defense against *F. pseudograminearum*.

Fig.5 The effect of phenazine on the antagonistic activity. (a): The inhibition zone diameters of wild-type, $\Delta phzF$ mutant strains and $phzF$ complemented strains against *F. pseudograminearum* on PDA plates; (e) and control efficacy index (f) of SR9 and its derivatives against *F. pseudograminearum* under greenhouse potted conditions;

The construction method and primer information of the complement strain have been added to the Methods (Line 382-388) and Supplementary sections (Table S4), and the new pictures have been updated.

Reviewers 2:

- *Fusarium pseudograminearum* is a soil-borne fungus that cause of wheat crown rot (WCR). The biocontrol assays of this disease are carried out by inoculating *P. khavaziana* SR9 and the pathogen in the soil. And the plant growth promotion activity is carried out root-irrigated with SR9. Why are potential biocontrol strains sought in stem samples and not in soil or roots samples?

Thank you for your question. While *Fusarium pseudograminearum* is a soil-borne pathogen causing wheat crown rot (WCR), the infection manifests primarily in the stem region. Consequently, the interception of the pathogen's invasion at the stem is as critical as controlling its soil transmission. It is

imperative to consider that the above-ground parts harbor unique microbial communities that are integral to plant health and defense mechanisms. And, in the pursuit of biological control strategies for wheat crown rot disease, our research has encompassed antagonist screening across rhizosphere soil, root, and stem tissues. Notably, the antagonistic strain SR9 was isolated from the stem tissue of healthy wheat plants.

We hypothesize that the isolation of SR9 from the stem may be attributed to several factors: Firstly, SR9, as an endophytic antagonist, establishes a symbiotic relationship with its host plant. This association enables the formation of an intrinsic biological defense barrier within the plant, offering protection against pathogen invasion. Secondly, the stem serves as a primary conduit for pathogen dissemination and disease manifestation, making antagonistic strains isolated from this region particularly adept at intercepting the disease transmission pathway. Lastly, the microbial communities within the stem environment have adapted to direct interactions with plant tissues, potentially enhancing their biocontrol efficacy. Thus, while soil and roots are conventional sources for antagonist screening, stem tissue-derived strains may also harbor significant potential for biological disease management.

- In materials and methods, you do not mention the Ashby's medium, nor the media used in Figure 6, please include a section about this in materials and methods

Thank you for your valuable feedback. We have revised the Materials and Methods section to include a detailed description of the media used. Specifically, we have clarified the composition of the Ashby's medium. The revised text, now found on lines 295-298.

- Erroneous assays that do not allow reliable results to be obtained, for example the figure 1, the distance of inoculation of the bacteria against the different pathogens are not at the same distance, I recommend repeating the antagonism assay as in Figure 5a.

Thank you for your meticulous review and the issues you've raised regarding figure 1. We acknowledge the oversight in the inoculation distances of the bacteria against the different pathogens. We have taken your

recommendation seriously and have repeated the antagonism assays with uniform distances. (Bacterial inoculum was spotted 2.5 cm away from the fungal disk.) The manuscript has been updated to reflect the new experimental results, and figure 1 has been revised accordingly. We appreciate your attention to detail and are grateful for the opportunity to enhance the accuracy of our work. Thank you once again for your valuable feedback.

Fig.1 Broad-spectrum antibacterial activity. Plate confrontation experiments and antimicrobial activity assessment of strain SR9 against *F. pseudograminearum*, *N. dictyophora*, *B. californica*, and *B. dothidea*. Data are mean \pm SD from three biological replicates.

- Lack of controls in the assays, e.g., in Figure 6 at least one negative control (bacteria with no activity) and one positive control (bacteria with assayed activity) must be used for each of the assays. The same for the biocontrol assays, no uninoculated plant controls and a positive control (another bacterium with known biocontrol in the pathosystem) are not applied.

Thank you for your critical insights regarding the control setups in our assays. Following your advice, we have now incorporated both negative and positive controls into our experimental design. The negative controls include treatments with no bacterial inoculation, inactivated SR9 strain, and/or

bacterial strains lacking functional activity. As for the positive controls, we have introduced treatments with *Pseudomonas fluorescens* strain 2P24, which is a well-documented biocontrol agent, or other bacterial strains with verified functional activity.

These enhancements have been applied to a range of assays, including those for the solubilization of inorganic and organic phosphates, siderophore production, nitrogen fixation, and the promotion of growth in wheat and *Arabidopsis* (Fig S7 and Fig 6). We believe these additions will provide a more robust validation of our findings.

Fig. S7 Functional characteristics of strain SR9. Solubilization of organic and inorganic phosphorus, production of siderophores and nitrogen fixation of strain SR9. 2P24 represents *P. fluorescens* 2P24; SR9-IA represents the prepared strain SR9 suspension was inactivated; Pch1 represents *Sinorhizobium fredii* Pch1; TG1 represents *Escherichia coli* TG1; and 88 represents *Ruicaihuangia caeni* 88.

Fig.6 Strain SR9 can produce IAA to promote plant growth. (a, b): Strain SR9 produces IAA and its quantification; CK+ represents 50 mg/L IAA, and CK- represents no bacteria suspension. Strain SR9 promotes wheat growth (c) and plant height difference (d), biomass difference (e, f); Strain SR9 can shorten the main root length (i) and increase the number of lateral roots (h) in *Arabidopsis thaliana* (g). 2P24 represents *P. fluorescens* 2P24; SR9-IA represents the prepared strain SR9 suspension was inactivated.

-In my opinion, determining the species *Pseudomonas khavaziana* is risky, I suggest to put *Pseudomonas spp.*, complementary tests should be performed for this determination.

We extend our gratitude for your valuable feedback. Advancing our research, we have executed an exhaustive comparative genomic analysis of strain SR9 vis-à-vis all accessible type strains within the *Pseudomonas* genus. This rigorous analysis incorporated Average Nucleotide Identity (ANI) and Digital DNA-DNA Hybridization (dDDH) assessments for 340 type strains, in conjunction with the genome of strain SR9. The outcomes revealed that strain SR9 manifests the most elevated ANI and dDDH values when compared with *Pseudomonas khavaziana* SWRI124 (Sup Data1 and Fig. S2).

Fig. S2 Average Nucleotide Identity analysis of related strains to SR9. FastANI was used to determine the average nucleotide identity of selected species within the genus *Pseudomonas*. GraphPad Prism 9 is used for visualization. 1,A: SR9; 2,B: *P.khavaziana* SWRI124^T; 3,C: *P.synxantha* DSM 18928^T; 4,D: *P.libanensis* DSM 17149^T; 5,E: *P.haemolytica* DSM 108987^T; The following colors represent the ANI comparison of strain SR9 with other different type strains. (For more, refer the Sup Data1)

Subsequently, we have delineated a phylogenomic tree featuring the top 40 type strains, which demonstrates that strain SR9 is phylogenetically congruent with *Pseudomonas khavaziana* SWRI124 (Fig. 3). Based on these compelling

results, we advocate for the classification of strain SR9 as *Pseudomonas khavaziana*. The revised content of this part has also been supplemented in the main manuscript. (Line106-110)

Fig.3 Phylogenetic status of strain SR9. Phylogenomic tree based on genome sequences, showing the phylogenetic positions of strain SR9 among closely related taxa. GenBank accession numbers are given in parentheses. *Komagataeibacter medellinensis* NBRC 3288^T were used as outgroups. ^T represents type strain.

-Some figures are out of order, e.g., Figure 7, letters are out of order both in figure legend and in the figure.

Thank you for your meticulous review. In response to your comments regarding the order of letters in Figure 7, we have conducted a thorough revision. We have now corrected the sequence of letters in both the figure legend and within the figure itself to ensure consistency with the text. Following on the subsequent comments, we have renumbered the original Figure 7 as the current Figure 6 (please refer to Figure 6 in light of these corrections). We appreciate your attention to detail, which has significantly contributed to the enhancement of our manuscript's quality.

Fig.6 Strain SR9 can produce IAA to promote plant growth.

-In my opinion, the swimming and biofilm tests do not add anything to the work.

Thank you for your insightful comments and suggestions regarding our manuscript. Upon reflection, we acknowledge that the discussion section may have initially lacked sufficient detail or reference to the pertinent literature, and we appreciate the opportunity to address this oversight.

Effective colonization of plant is a pivotal factor for the success of biocontrol bacteria. Motility, facilitated by mechanisms such as flagella-driven swimming and swarming, enables bacteria to navigate the rhizosphere and locate optimal colonization sites (1) (2). Once established, biofilm formation creates a protective microenvironment that enhances bacterial survival and persistence under fluctuating environmental conditions (3) (4), thereby promoting plant development, nutrient uptake, and pathogen suppression (5).

We have since supplemented the discussion to emphasize that the motility and biofilm formation capabilities of bacteria are indeed critical prerequisites for successful colonization of plant. Motility allows bacteria to reach and colonize the root zone, while the ability to form biofilms enables them to establish stable communities on plant surfaces and roots. It is only upon successful colonization that bacteria can effectively exert their biocontrol and plant growth-promoting functions.

Therefore, we maintain that tests for motility and biofilm formation are necessary to fully understand the mechanisms underlying the biocontrol efficacy of these bacteria.

-The discussion consists of few references (7), I think that much of it should be rewritten, focusing on discussing the wide variety of mechanisms and potential characteristics that *Pseudomonas khavaziana* strain SR9 has to be a good biocontrol and plant growth-promoting bacterium, and fails arising up relevant results.

We are grateful for your insightful comments and recommendations. The discussion section has been thoroughly revised to address the issues you have raised. The enhanced discourse now probes the multifarious

mechanisms and characteristics of the *P. khavaziana* strain SR9, with a focus on:

1. Biocontrol Agent Diversity: Highlighting the diversity of effective biocontrol agents and the antagonistic action of SR9 against *F. pseudograminearum*.
2. Colonization and Biofilm Formation: Analyzing the significance of motility and biofilm formation for successful colonization, supported by genomic data.
3. Phenazine Biosynthesis and Regulatory Mechanisms: Investigating the antagonism properties of phenazines, their biosynthetic pathways, and the regulatory mechanisms involving the GacS/GacA two-component system.
4. Nutrient Supply and Symbiotic Enhancement: Detailing SR9's role in phosphate solubilization, siderophore production, nitrogen regulation, and IAA synthesis, including specific gene functions and empirical validations.
5. Sustainable Agriculture Outlook: Expanding upon the strategic integration of SR9 within diverse agricultural frameworks, particularly in rotations involving cruciferous and gramineous species.

We trust that these revisions will satisfactorily resolve the highlighted concerns. We extend our gratitude for your time and constructive feedback, which have been pivotal in enhancing our manuscript.

-Figure 1, in the case of antagonism against *N.dictyophora* and *B.dothidea*, the growth of the fungus is irregular. How did you measure the diameter? I think that the result shown in the graph is not representative of the image. SR9 strain inhibit *F. pseudograminearum* better than *B.dothidea*, however the growth measurement is similar. I suggest that it would be more correct in this case to measure the growth area of the fungus.

Thank you for your insightful comments, which are invaluable for the refinement of our manuscript. Upon reevaluation, we agree that quantifying the mycelial growth area is a more precise approach to assess the antagonistic effects of SR9 against the pathogenic fungi. To address this, we have conducted additional experiments and utilized ImageJ software to calculate both the mycelial growth area and the corresponding inhibition rates.

These revised measurements provide a more accurate representation of the SR9 strain's efficacy. We extend our sincerest gratitude for your constructive critique and have amended our manuscript accordingly to incorporate these enhanced methodological details. (Line 312-313 and Figure 1)

Fig.1 Broad-spectrum antibacterial activity. Plate confrontation experiments and antimicrobial activity assessment of strain SR9 against *F. pseudograminearum*, *N. dictyophora*, *B. californica*, and *B. dothidea*. Data are mean \pm SD from three biological replicates.

-Line 91, Please, change “antifungal activity” to “antagonism or inhibition of growth”, the fungal inhibition growth could be due to different factors, not only the production of antifungal compounds.

Thank you for your meticulous review. We acknowledge the complexity of fungal growth inhibition and agree that it can result from various factors beyond the production of antifungal compounds. Accordingly, we have revised the manuscript to replace “antifungal activity” with “antagonism” to more accurately reflect the range of possible inhibitory mechanisms. We appreciate your attention to detail and have made the necessary corrections throughout the manuscript. (Line29, 34, 90, 92, 304, 306, 313)

-Line 171, eliminate the word "infection". In assays of Fig. 5a, only growth inhibition is observed against *F.pseudograminearum*, not infection.

We are thankful for your detailed review and the specific observation regarding line 170. In accordance with your suggestion, we have removed the word 'infection' to more accurately describe the observations from Fig. 5a, which indeed reflect growth inhibition of *F. pseudograminearum* rather than infection. We appreciate your guidance in enhancing the precision of our manuscript.

-Line 190-195, I suggest changing this paragraph from materials and methods to discussion, to discuss the importance of participation in the nitrogen and sulphur cycle for potential bacterial biocontrol and plant growth promotion. The same for line 201-203.

We extend our gratitude for your valuable feedback and guidance. In accordance with your directives, we have conducted a comprehensive rewrite of the discussion section.

-Line 303, The use of the word "Biocontrol" should be replaced by "antagonism", since the screening is with a plate antagonism assay.

We thank you for your astute observation regarding near line 303. Following your recommendation, we have replaced the word 'Biocontrol' with 'Antagonism' to more accurately reflect the methodology employed, namely the plate antagonism assay. This change ensures that the terminology used is consistent with the experimental approach. We appreciate your attention to detail and have made the amendment as suggested. (Line 259, 269)

-Line 306, Please, indicate the number of isolates that are tested in the screening

Thank you for your suggestion. The manuscript has been updated to include the specific number of bacterial strains screened. A total of 37 endophytic bacterial strains were isolated from wheat stems for the screening process. (Line 277)

-Figure S1, I suggest that at least the growth curve of the SR9 strain in LB include bacterial counts (cfu/ml), not just optical density measurements.

Thank you for your valuable suggestion regarding the inclusion of bacterial counts alongside optical density measurements for the growth curve of the

SR9 strain in LB medium. We have taken your advice into consideration and have re-evaluated the growth curve of SR9, incorporating both optical density (OD₆₀₀) measurements and colony-forming unit (CFU/ml) counts. This dual parameter assessment will provide a more comprehensive understanding of the bacterial growth dynamics (Fig S1b).

Fig. S1 Carbon and nitrogen source utilization, colony morphology, and growth curve analysis of strain SR9. (b): Growth curve analysis of strain SR9 in LB medium. The OD₆₀₀ and colony counts were measured at different time points, The squares represent the colony counts of strain SR9 at different time points, while the circles indicate the optical density (OD₆₀₀) readings at those same time points.

-Figure 3, Although the comparison of the genome of the SR9 strain and the SWRI124^T strain of *P. khavaziana* confirmed their high similarity, with an average nucleotide identity of 97.73%, I suggest that more strains be included in the phylogenetic tree.

We are thankful for your insightful suggestions. In response, we have augmented our analysis by including the top 40 type strains of the *Pseudomonas* genus, exhibiting high ANI and dDDH similarities, to construct a comprehensive phylogenomic tree. The results consistently demonstrate that strain SR9 clusters with *P. khavaziana* SWRI124^T. We have updated the corresponding figure to reflect these findings (Fig. 3).

Fig.3 Phylogenetic status of strain SR9. Phylogenomic tree based on genome sequences, showing the phylogenetic positions of strain SR9 among closely related taxa. GenBank accession numbers are given in parentheses. *Komagataeibacter medellinensis* NBRC 3288^T were used as outgroups. ^T represents type strain.

-Figure 5d, I suggest including, along with the photos, a graph with the disease control data, to observe if there are significant differences between

the *phzF* mutant and the wild strain. It may include disease index at the end of the trial, or the area of progress under the disease curve.

Thank you for your valuable suggestion. In response to your recommendation, we have not only supplemented our manuscript with the disease index and efficacy index of the SR9 strain and its derivatives during the seedling stage (the measurement of wheat disease index where we selected 4 plants per treatment group for three independent experiments) (Fig. 5g-h) but also included these indices under greenhouse pot conditions (the measurement of wheat disease index where we selected 10 plants per pot with 6 pots per treatment, resulting in a total of 60 plants per treatment group for three independent experiments) (Fig. 5e-f)

Fig.5 The effect of phenazine on the antagonistic activity. Disease index (e) and control efficacy index (f) of SR9 and its derivatives against *F. pseudograminearum* under greenhouse potted conditions; Disease index (g) and control efficacy index (h) of SR9 and its derivatives against *F. pseudograminearum* in greenhouse seedling stage.

-Line 327: Please, change “Antifungal Activity Assay” to “Antagonism Assay”, With this assay, the antagonistic capacity against the pathogen is observed, not the antifungal activity. The same for line 329 and 335.

We are grateful for your review. The manuscript has been amended. (Line 304, 306, and 313)

-Line 337: You mention that the inhibition ratio was calculated, however in the graph of Figure 1 show mycelium diameter in cm. Change it in materials and methods.

We appreciate your attention to the details in our manuscript. As per your observation, we have revised the 'Materials and Methods' section (Line 313-315) and figure information to reflect the measurement of mycelium area, which corresponds with the data presented in Figure 1. The calculation of inhibition rate corresponds to the information in the manuscript (Line 89 and Line 315). Additionally, we have conducted a thorough review of the entire document to ensure consistency between the methods described and the results displayed. We are grateful for your guidance, which has helped improve the clarity and accuracy of our work.

-Figure 7g, please, show a closer photo, to observe the increase in the lateral roots.

Thank you for your suggestion to provide a closer view of the lateral roots in Figure 7g. We have taken your advice into consideration and have supplemented our manuscript with a high-resolution micrograph of the Arabidopsis lateral roots (now presented as Figure 6g).

Fig.6 Strain SR9 can produce IAA to promote plant growth. (a, b): Strain SR9 produces IAA and its quantification; CK+ represents 50 mg/L IAA, and CK- represents no bacteria

suspension. Strain SR9 promotes wheat growth (c) and plant height difference (d), biomass difference (e, f); Strain SR9 can shorten the main root length (i) and increase the number of lateral roots (h) in *Arabidopsis thaliana* (g). 2P24 represents *P. fluorescens* 2P24; SR9-IA represents the prepared strain SR9 suspension was inactivated.

-Figure 7b, e and f, I suggest using the same type of graph as in 7d, h and I, showing with dots, all the replicas.

Thank you for your constructive comments regarding the graphical representation in Figure 7b, e, and f. We have standardized our statistical figures to maintain uniformity across the manuscript (now presented as Figure 6). We have now adopted bar graphs for these figures. Due to the varying quantities associated with each experiment, such as the measurement of wheat plant height where we selected 10 plants per pot with 6 pots per treatment, resulting in a total of 60 plants per treatment group, the data representation needed to be adapted accordingly. For the dry and fresh weight statistics, each pot of 10 plants was considered a replicate, yielding four data points per treatment group. The adoption of bar graphs serves to present the data in a unified and aesthetically pleasing manner. We hope this meets your approval and enhances the clarity of our results presentation.

-Line 443-445, "Each seedling was inoculated with 10 µl of strain SR9 suspension (1×10^8 CFU/ml) at 5 cm from the base, while the control seedlings were treated with 10 µl of sterile PBS". Bacterial suspension in what is it? in culture medium? sterile PBS? I think it is important that you indicate this.

We appreciate your careful reading and the request for clarification. In response to your query, we confirm that the bacterial suspension of strain SR9 was prepared in sterile phosphate-buffered saline (PBS). The seedlings were inoculated with 10 µl of this SR9 PBS suspension, which had a concentration of 1×10^8 CFU/ml. This detail was indeed mentioned in the 'Characterization of strain SR9' section : harvested by centrifugation and resuspended in sterile phosphate-buffered saline (PBS) to achieve a final concentration of 1×10^8 CFU/ml for further characterization and experimental procedures.

However, we realize it may not have been sufficiently clear. We have now amended the manuscript to explicitly state that the SR9 suspension used for inoculation was prepared in sterile PBS. Thank you for bringing this to our attention, ensuring the accuracy and clarity of our methods. (Line 326, 342, 345, 418, and 427)

-Line 556, I suggest to change the name of this section by biofilm adhesion.

We thank you for your recommendation and have made comprehensive revisions throughout this section in accordance with your advice. (Line 35, 433, 434, and 795)

1. de Weert S, Vermeiren H, Mulders IHM, Kuiper I, Hendrickx N, Bloemberg GV, Vanderleyden J, De Mot R, Lugtenberg BJJ. 2002. Flagella-Driven Chemotaxis Towards Exudate Components Is an Important Trait for Tomato Root Colonization by *Pseudomonas fluorescens*. *Mol Plant Microbe In* 15:1173-1180.
2. Ramoneda J, Fan K, Lucas JM, Chu H, Bissett A, Strickland MS, Fierer N. 2024. Ecological relevance of flagellar motility in soil bacterial communities. *The ISME Journal* 18(1): wrae067.
3. Ajjah N, Fiodor A, Pandey AK, Rana A, Pranaw K. 2023. Plant Growth-Promoting Bacteria (PGPB) with Biofilm-Forming Ability: A Multifaceted Agent for Sustainable Agriculture. *Diversity* 15:112.
4. Li Y, Narayanan M, Shi X, Chen X, Li Z, Ma Y. 2024. Biofilms formation in plant growth-promoting bacteria for alleviating agro-environmental stress. *Sci Total Environ* 907:167774.
5. Ünal Turhan E, Erginkaya Z, Korukluoğlu M, Konuray G. 2019. Beneficial Biofilm Applications in Food and Agricultural Industry. In Malik A, Erginkaya Z, Erten H (eds), *Health and Safety Aspects of Food Processing Technologies*. Springer, Cham.

Re: Spectrum00712-24R1 (Unveiling the multifaceted potential of *Pseudomonas khavaziana* strain SR9: a promising biocontrol agent for Wheat Crown Rot)

Dear Prof. Yao Wang:

The comments and observations made by the reviewers have been attended and answers have been provided in great detail. I am now pleased to be able to accept your manuscript for publication in Microbiology Spectrum.

Your manuscript has been accepted, and I am forwarding it to the ASM production staff for publication. Your paper will first be checked to make sure all elements meet the technical requirements. ASM staff will contact you if anything needs to be revised before copyediting and production can begin. Otherwise, you will be notified when your proofs are ready to be viewed.

Sincerely,
Frédérique Reverchon
Editor
Microbiology Spectrum